# Antioxidant, Anti-Inflammatory, and Antiapoptotic Effects of *Euterpe oleracea* Mart. (Açaí) in Improving Cognition Deficits: Potential Therapeutic Implications for Alzheimer’s Disease

**DOI:** 10.3390/plants14132010

**Published:** 2025-06-30

**Authors:** Flávia dos Santos Ferreira, Juliana Lucena Azevedo de Mattos, Paula Hosana Fernandes da Silva, Cristiane Aguiar da Costa, Dayane Teixeira Ognibene, Angela de Castro Resende, Graziele Freitas de Bem

**Affiliations:** 1Department of Pharmacology and Psychobiology, Roberto Alcantara Gomes Biology Institute (IBRAG), Rio de Janeiro State University (UERJ), Rio de Janeiro 20551-030, RJ, Brazil; santosflavia887@gmail.com (F.d.S.F.); julianazevedomt00@gmail.com (J.L.A.d.M.); crysac84@yahoo.com.br (C.A.d.C.); dayognibene@gmail.com (D.T.O.); angelacr@hotmail.com (A.d.C.R.); 2Department of Pharmacology, Institute of Biomedical Science, University of Sao Paulo (USP), São Paulo 05508-000, SP, Brazil; paulahosana00@gmail.com

**Keywords:** *Euterpe oleracea*, açaí pulp, açaí seed, neuroprotection, Alzheimer’s disease

## Abstract

*Euterpe oleracea Martius*, also popularly known as açaí palm, is a palm tree of the *Aracaceae* family widely found in the Amazon region. Traditional plant use reports indicate the beneficial effects of açaí juice on fever, pain, and flu. Moreover, many studies have demonstrated the pharmacological potential of açaí, mainly the pulp and seed of the fruit, due to its chemical composition, which significantly consists of polyphenols. In recent years, there has been a growing interest in investigating the neuroprotective effects of açaí, with the potential for the prevention and treatment of neurodegenerative diseases, such as Alzheimer’s disease, mainly due to the increasing aging of the population that has contributed to the increase in the number of individuals affected by this disease that has no cure. Therefore, this review aims to evaluate the potential role of açaí fruit in preventing or treating cognitive deficits, highlighting its potential in Alzheimer’s disease therapy. Preclinical in vivo and in vitro pharmacological studies were utilized to investigate the learning and memory effects of the pulp and seed of the açaí fruit, focusing on antioxidant, anti-inflammatory, antiapoptotic, and autophagy restoration actions.

## 1. Introduction

Previous studies demonstrated that polyphenols derived from fruits and vegetables promoted a lower incidence of neurodegenerative diseases [1,2,3]. They can modulate different cell signaling pathways [4], including nerve cells, by influencing neuronal survival, regeneration, development, or death [5]. Additionally, polyphenols have a powerful antioxidant and anti-inflammatory action, which is involved in their neuroprotection actions [6]. Moreover, research on medicinal plants’ neuroprotective effects has gained significant prominence, as the prevalence of neurodegenerative diseases has increased with the population’s life expectancy.

Neurodegenerative disorders such as Alzheimer’s, Parkinson’s, Huntington’s, and Lateral Amyotrophic Sclerosis diseases are characterized by extracellular protein deposits, cellular inclusions, changes in cell morphology, and progressive and irreversible loss of neurons in specific brain regions, resulting in functional and mental impairment [7,8]. Moreover, this neuronal degeneration progressively diminishes essential body activities, such as movement, coordination, breathing, balance, speech, and the functioning of vital organs [9].

Among neurodegenerative diseases, Alzheimer’s disease (AD) is the most prevalent in the world population and the primary cause of dependency and disability in elderly individuals [10,11]. This disease is responsible for 60% to 80% of dementia cases in elderly individuals [12,13]. AD dementia is a specific form of cognitive and functional decline associated with age and late-onset due to molecular alterations that can appear up to 20 years before the first symptoms, with interference in episodic memory (amnesia) being the earliest and most apparent factor [14,15].

Notably, dementia caused more than one million deaths worldwide, becoming the seventh leading cause of death in 2019. Moreover, this syndrome affected around 55 million people worldwide in 2019, and it is estimated that this number will double every 20 years [13]. Therefore, in the year 2050, this number is expected to increase to around 139 million cases [16]. In Brazil, according to the Ministry of Health, approximately 1.2 million people have some dementia type, and there are 100,000 new cases each year. Furthermore, between 2007 and 2017, dementia deaths increased by 55% among Brazilians. According to Paschalidis and collaborators (2023) [16], from 2000 to 2019, 211,658 deaths were recorded among Brazilians due to AD, and 64% of these individuals were women.

AD is a progressive neurodegenerative disease with a silent onset [17]. Its main pathophysiological characteristics are the presence of beta-amyloid plaques in the extracellular space, neurofibrillary tangles of hyperphosphorylated tau protein in the intracellular environment, and an elevated neuron loss in specific central nervous system regions. Moreover, there is an acetylcholine (ACh) level reduction, synaptic loss, and cholinergic neuron death in the cerebral cortex, hippocampus, entorhinal cortex, and ventral striatum, compromising cognitive functions [18]. At a macroscopic level, there is notable tissue atrophy in regions such as the cortex and hippocampus [19]. The mechanisms triggering cell death and synaptic damage in AD might be related to inflammation [20,21] and oxidative stress [22,23], in which polyphenols from medicinal plants play a prominent role.

The most widely used therapeutic basis for treating AD is the amplification of cholinergic transmission with reversible cholinesterase inhibitors, which are the primary symptomatic treatment for the cognitive deficits that occur in AD [24]. In June 2021, after 18 years without new treatments for this pathology, the use of Aducanumab (a human monoclonal antibody) was approved by the Food and Drug Administration. The new drug is intended for the treatment of AD in the phase of mild cognitive impairment and mild dementia, targeting the beta-amyloid protein. However, researchers have argued that the relevance of the clinical findings is questionable because although the drug induces a reduction in the density of beta-amyloid plaques, the clinical response did not show significant interference in the performance and functionality of patients [25,26]. Therefore, the emergence of the drug is an important milestone, but further studies are necessary to confirm its clinical effects. Thus, these drugs only slow the progression of the disease since there is no cure, which causes great suffering for the patient and their family members, highlighting the importance of studying new therapeutic agents that can prevent or treat neurodegeneration and cognitive deficits. 

The incredible plant biodiversity of Brazil may represent a natural source of drugs, enabling the use of medicinal plants as an alternative therapeutic resource that has been growing in the medical community. Notably, recent data have highlighted that among the various substances extracted from plants, polyphenols have demonstrated great therapeutic potential since epidemiological and preclinical studies suggest their properties in the treatment and prevention of neurodegeneration and neurotoxicity present in neurodegenerative diseases due to their antioxidant, anti-inflammatory, and antiapoptotic potential [27]. Regarding these promising polyphenol properties, *Euterpe oleracea Martius*, a palm tree from which açaí comes, native to Brazil, is a medicinal plant rich in polyphenols, which have potent antioxidant and anti-inflammatory action, demonstrating therapeutic potential for treating AD. Therefore, in this review, we intend to deepen our knowledge of the actions of *Euterpe oleracea* on cognitive deficits, inflammation, oxidative stress, neurogenesis, apoptosis, and autophagy through preclinical studies to highlight key mechanisms of this medicinal plant for the treatment of AD.

## 2. *Euterpe oleracea Martius*

### 2.1. Euterpe oleracea Martius Botanical Description

The plant *Euterpe oleracea Martius*, also popularly known as açaí palm (Figure 1), is a palm tree of the *Aracaceae* family, widely found in the Amazon region, in Brazilian states such as Pará, Amazonas, Tocantins, Maranhão, and Amapá [28]. This plant is also native to Ecuador and Venezuela [29]. In the Brazilian Amazon, flowers and fruits are found on the açaí tree all year round. However, in Pará, flowering occurs during the rainiest season (January to May) and fruiting during the driest periods (September to December) [30].

Moreover, the açaí palm is a caespitose palm tree, with up to 25 shoots per clump at different stages of development. Adult plants have stems measuring 3 to 20 m in height and 7 to 18 cm in diameter [31]. The leaves are compound, pinnate, and have a spiral arrangement of 40 to 80 pairs of leaflets. The cluster-type inflorescence has staminate and pistillate flowers [32,33]. Two male flowers flanked one female flower, arranged in triads. The fruit of the açaí palm is a globose drupe measuring 1 to 2 cm in diameter and weighing an average of 1.5 g. When ripe, the fruit epicarp can be purple or green during maturation [34]. The pulpy mesocarp (ca. 1 mm thick) surrounds the voluminous, hard endocarp that follows the shape of the fruit and contains the seed inside [35,36].

Although the consumption of açaí by Amazonian populations is ancient, it is only in the 21st century that this food product has attracted the interest of markets outside the region, both nationally and internationally. However, there is a marked difference in consumption patterns. In the Amazon, açaí is consumed in meals as a main food and is served with fish or meat and flour. Outside this region, it is considered an energy drink mixed with sugar and other products such as guarana syrup, granola, banana, peanuts, and condensed milk [37]. Given the widespread use of the *Euterpe oleracea* (açaí) fruit as a functional food and its significant polyphenolic content, it has attracted the attention of scientists and even more so of national and foreign industries that import tons of this fruit from the Amazon region for industrialization and research development, mainly in the United States, China, and Japan [38].

The use of açaí in food, as a dietary supplement, and in scientific research has led to an enormous global demand for the fruit, making Brazil stand out and emerge as the largest producer and exporter [32]. The traditional plant use reports, mainly among people from the north and northeast regions, indicate the beneficial effects of açaí juice for fever, pain, and flu [39]. Moreover, many studies have demonstrated the pharmacological potential of açaí, mainly the pulp and seed of the fruit, due to its chemical composition, significantly consisting of polyphenols [36].

### 2.2. Açaí Pulp Chemical Composition and Pharmacological Actions

Chemical studies have shown that açaí pulp is mainly composed of quercetin, (+)-catechin, cyanidin-3-glucoside, vanillic acid, cyanidin-3-rutinoside, p-hydroxybenzoic acid, ferulic acid, protocatechuic acid, and syringic acid (Figure 2) [40,41]. These bioactive compounds are responsible for a variety of pharmacological properties. Supplementation with açaí pulp promotes beneficial effects on cardiometabolic changes. In this regard, the literature investigations have demonstrated that *Euterpe oleracea* pulp decreases hyperglycemia in rats subjected to streptozotocin [42].

Previous studies have shown that açaí pulp reduces dyslipidemia [43] by increasing the gene expression of the cholesterol transporters and ATP-binding cassette sub-family G members 5 and 8 (ABCG5 and ABCG8) in rats on a hypercholesterolemic diet [44]. Açaí pulp also improves hepatic steatosis through the increased expression of paraoxonase one in high-fat-diet-fed rats [45]. Moreover, this fruit portion mitigates atherosclerosis in APOE-deficient mice [46] and cardiac remodeling in rats subjected to myocardial infarction [47]. It also reduces cardiac hypertrophy and cardiomyocyte contractility in high-fat-diet-fed rats [48]. Additionally, it elevates acute blood flow with nitric oxide (NO) involvement in healthy rats [49].

In addition, açaí pulp has demonstrated antimicrobial actions against *Staphylococcus aureus*, acting synergistically with other antimicrobial drugs [50]. Its antitumor properties are also noteworthy, as it reduces tumor cell proliferation and dysplasia in colon cancer cells [51,52]. Furthermore, the fruit pulp decreases tumor size, mitosis, and pleomorphism while increasing tumor necrosis in solid Ehrlich tumors in mice [53]. Previous data have also shown that açaí pulp decreases transitional cell carcinoma, p63 expression, and tumor cell proliferation in urothelial bladder carcinogenesis in mice [54].

Studies have also investigated the beneficial effects of açaí pulp in humans [55,56,57,58,59,60,61,62]. Data from the literature have shown that consumption of fruit pulp for fifteen days increased antioxidant capacity and reduced serum lipid peroxidation. It also decreased lactate levels during physical exertion and increased the intensity of the anaerobic threshold in male cyclists [55]. Supplementation with açaí pulp for four weeks also reduced the production of reactive species, increased total antioxidant capacity [56], and decreased serum levels of visfatin, leptin, and P-selectin in healthy women [60]. In addition, it increased paraoxonase one antioxidant activity and enhanced cholesteryl ester transfer to HDL and apolipoprotein A-I (APOA-I) concentrations, suggesting the potential of açaí pulp against atherosclerosis [61]. A previous study also demonstrated that combining a hypocaloric diet with açaí pulp supplementation for sixty days improved inflammation and decreased oxidative stress in patients with overweight and dyslipidemia [59]. Finally, *Euterpe oleracea* pulp supplementation for one month reduced fasting glycemic, insulinemic, and lipid profile levels in overweight individuals [62].

### 2.3. Açaí Seed Chemical Composition and Pharmacological Actions

The açaí seed accounts for approximately 80% of the fruit (Figure 1), which can weigh between 0.6 and 2.8 g and have a diameter of 0.6 to 2.5 cm [36]; seeds discarded generate tons of waste and have a significant environmental impact. Previous studies have demonstrated that *Euterpe oleracea* seeds are rich in polyphenols, such as catechins, epicatechins, and polymeric proanthocyanidins (Figure 3) [63,64], which exhibit numerous pharmacological properties related to cardiometabolic changes.

Açaí seeds prevent the development of hypertension, endothelial dysfunction, and cardiovascular remodeling in spontaneously hypertensive rats [65], in a renovascular hypertension model [66,67], and obesity induced by a high-fat diet [68,69]. They also mitigate metabolic programming caused by protein restriction [70] and nitro-L-arginine methyl ester (L-NAME) administration during the gestational period [71]. These effects involve increased antioxidant activity [65,66,67,69,70], enhanced NO bioavailability in endothelial cells [28], decreased plasma renin levels [66,70], and modulation of the local renin–angiotensin system in adipose tissue [69]. In addition to its preventive effects, the seed reverses arterial hypertension and cardiovascular remodeling [72].

Previous data have also demonstrated the anti-obesity effects of *Euterpe oleracea* seed extract, particularly concerning obesity-related hyperglycemia and hyperinsulinemia [64,68,69,73]. Regarding these properties, its fruit component prevents the development of hepatic steatosis and dyslipidemia through elevated cholesterol excretion and decreased lipogenesis in obesity induced by a high-fat diet [64,73]. Moreover, the açaí seed prevents adipocyte hypertrophy and activates the local renin–angiotensin system in adipose tissue [69]. Once established, the açaí seed treats obesity and steatosis [74,75].

Other studies have demonstrated its antidiabetic effects in a model of type 2 diabetes induced by a combination of a high-fat diet and a low dose of streptozotocin [76,77]. The pharmacological actions of the açaí seed involve increasing glucagon-like peptide-1 (GLP-1) levels and activating the insulin signaling pathway in the adipose tissue and skeletal muscle [77], as well as elevating AMP-activated protein kinase (pAMPK) expression in the liver [76], which contributes to increased glucose uptake and reduced glycemic levels. Furthermore, in a model of type 1 diabetes induced by streptozotocin, the seed reduces renal fibrosis [78].

It is worth noting that the *Euterpe oleracea* seed also increases the distance covered and the time spent exercising on a treadmill in both healthy adults [79] and elderly rats [80] through the regulation of mitochondrial biogenesis, antioxidant action, and improvements in vascular function [79,80]. Additionally, a previous study demonstrated an antinociceptive effect of the açaí seed in acute and neuropathic pain rat models [81]. Therefore, this versatility in treating various health conditions underscores the potential impact of açaí in the health and nutrition field.

### 2.4. Açaí Pulp and Seed Antioxidant and Anti-Inflammatory Actions in Peripheral Tissues: Perspective for AD Treatment

The antioxidant and anti-inflammatory properties of medicinal plants rich in polyphenols, such as *Euterpe oleracea*, play a prominent role in the beneficial pharmacological effects they provide. In this context, numerous studies have demonstrated the antioxidant effects of açaí pulp [41,43,45,47,48,54,56,57,59,61,82,83,84,85,86] and açaí seeds [64,65,66,67,68,69,70,71,72,73,74,75,76,77,78,79,80,87,88,89,90]. These studies also highlight their anti-inflammatory effects on peripheral tissues in various experimental models [59,63,69,72,73,77,78,83,88,90,91,92].

The observation of these beneficial antioxidant and anti-inflammatory actions, combined with the potential of açaí pulp and seeds to promote effects on the central nervous system, evidenced by their anxiolytic [89,93] and anticonvulsant properties [94], highlights the possibility of using *Euterpe oleracea* in the prevention and treatment of neurotoxicity and neurodegeneration present in AD.

## 3. *Euterpe oleracea Martius* Actions on the Central Nervous System

### 3.1. Euterpe oleracea and Improved Cognition

Cognitive and memory deficits are the symptoms of AD, and the presence of beta-amyloid plaques confirms the progression of the disease. As a disorder with a complex pathophysiology for which there is no cure, many studies have investigated new strategies and therapeutic targets, demonstrating the high potential of medicinal plants, such as *Euterpe oleracea*.

A previous study demonstrated that *Euterpe oleracea* supplementation increases spatial memory retention in Wistar rats subjected to scopolamine and mecamylamine administration in the Morris water maze. In this behavioral test, açaí, at doses of 100 mg/kg and 300 mg/kg, increased the time spent in the platform quadrant, as observed in animals treated with rivastigmine, a drug used to slow the progression of memory deficits [95]. In this context, the hippocampus is critical in forming, organizing, and storing new memories [96,97,98]. In AD, there is cholinergic dysregulation between the basal forebrain and its target tissues, such as the hippocampus, resulting from the neurodegeneration of neurons that synthesize acetylcholine and contributing to the development of memory deficits [99,100]. Notably, açaí fruit increased hippocampal ACh concentrations in rats subjected to scopolamine and mecamylamine administration, contributing to memory improvement (Table 1 and Figure 4). Therefore, the memory enhancement induced by *Euterpe oleracea* appears to involve nicotinic and muscarinic cholinergic signaling pathways [95]. Moreover, a recent study has demonstrated that the aqueous and ethanolic extracts of açaí pulp inhibit the activity of acetylcholinesterase and butyrylcholinesterase [101]. This effect plays a prominent role in the potential of *Euterpe oleracea* in the treatment of AD, as the primary treatment available for this condition is the use of cholinesterase inhibitors to increase cholinergic neurotransmission [102]. Additionally, there is evidence demonstrating the expression of butyrylcholinesterase in the brains of patients with AD [103].

Supplementation with açaí pulp also reduced the latency to find the platform in elderly rats in the Morris water maze, thereby improving reference and spatial memory (Table 1 and Figure 4). The authors suggested that this effect on the fruit may involve reduced microglial activation and NO levels induced by the polyphenols in their chemical composition [104]. Another study demonstrated that açaí pulp improves learning and memory in obese rats subjected to a high-fat diet in the object recognition test by increasing the time spent with the novel object (Table 1 and Figure 4) [105]. Additionally, a previous finding highlights that obesity and diabetes contribute to cognitive impairment and AD development [115]. Thus, the fruit restores insulin sensitivity and elevates adiponectin levels and antioxidant activity, contributing to cognition [105].

Furthermore, *Euterpe oleracea* fruit pulp mitigates learning and memory deficits in rats subjected to vascular dementia in the object recognition test by increasing the time spent with the novel object. Notably, açaí decreases hippocampal neuronal death by reducing neurotoxicity and neurodegeneration (Table 1 and Figure 4). These actions involve its antioxidant and antiapoptotic properties and restore autophagic flux in neurons within the hippocampus [106]. Finally, treatment with açaí extract mitigated the deformation of red blood cells subjected to D-galactose and collected from healthy volunteers aged 20 to 60 years, a model of aging [111]. Another investigation demonstrated that açaíextract prevented the formation of acanthocytes and leptocytes in erythrocytes subjected to D-galactose [112]. In patients with AD, there is a reduction in these cells and changes in their morphology, which are related to an increased risk of developing dementia and cognitive impairment [116]. These findings may be promising for the treatment of AD, as these changes occur with aging and contribute to the disease’s pathophysiology. Therefore, the cognitive improvement induced by *Euterpe oleracea* is related to its polyphenolic content, highlighting its antioxidant, anti-inflammatory, antiapoptotic, and autophagy-restorative effects, which may reduce neurotoxicity and neurodegeneration in AD.

### 3.2. Euterpe oleracea Antioxidant and Anti-Inflammatory Actions

Data from the literature have shown that chronic brain inflammation, mainly involving microglia, astrocytes, and neurons, is related to AD development. An exacerbated expression of pro-inflammatory mediators, such as interleukin 6 (IL-6) and tumor necrosis factor alpha (TNF-α), can characterize the neuroinflammation present in this neurodegenerative disease. There may be a relationship between the formation of senile plaques and pro-inflammatory cytokines, which would increase neurological damage [20]. The bioactive compounds of *Euterpe oleracea* have relevant anti-inflammatory action, which can effectively contribute to treating neurotoxicity and neurodegeneration present in AD. Extracts obtained from açaí pulp prepared with ethanol, methanol, ethyl acetate, and acetone demonstrated anti-inflammatory action in microglial cells of BV-2 mice with LPS-induced inflammation. In this study, the polyphenols present in the pulp extracts reduced NO release, cyclooxygenase-2 (COX-2) expression, nuclear factor kappa B (NFkB) phosphorylation, TNF-α levels, and inducible nitric oxide synthase (iNOS) production [40]. Another investigation demonstrated that a hydroalcoholic extract obtained from açaí pulp caused an inhibition of the NLRP3 inflammasome and a reduction in IL-1β in the microglial EOC cell line subjected to LPS and Nigericin [113]. In elderly animals fed açaí pulp, there was also a reduction in the expression of NFkB in the hippocampus, a transcription factor with a key role in inflammation [107]. Moreover, açaí pulp supplementation prevented the elevation of TNF-α, IL-1β, and IL-18 levels in the hippocampus, cortex, and cerebellum in rats submitted to intraperitoneal administration of carbon tetrachloride (Table 1 and Figure 4) [117]. Notably, a recent in silico study demonstrated that açaí chemical compounds promote their anti-inflammatory effect by inhibiting the NLRP3 inflammasome. Furthermore, this investigation highlighted the low toxicity of açaí compounds compared to MCC950 and NP3-146 (RM5), which are synthetic NLRP3 inhibitors [114], demonstrating their pharmacological potential and safety.

It is also noteworthy that this abnormal protein aggregation is one of the main characteristics of AD, which induces neuroinflammation and additionally causes mitochondrial dysfunction, promoting an exacerbated production of reactive oxygen and nitrogen species, whose excessive accumulation triggers oxidative stress and neuronal apoptosis [21,22,23]. Some evidence suggests that the dysregulation of amyloid precursor protein (APP) processing begins with this exacerbated production of reactive species, contributing to beta-amyloid plaque formation [22]. In this sense, *Euterpe oleracea* bioactive compounds also demonstrated potent antioxidant actions. Açaí pulp supplementation reduced lipid peroxidation in the cerebral cortex and cerebellum and protein carbonylation in the cerebral cortex, cerebellum, and hippocampus in rats exposed to carbon tetrachloride [117]. In addition, açaí increased the catalase antioxidant activity in the hippocampus and cerebellum and the superoxide dismutase activity in the hippocampus (Table 1 and Figure 4) [117]. Similar results were observed in the previously mentioned brain tissues of rats treated with açaí pulp and subsequently subjected to hydrogen peroxide (Table 1 and Figure 4), demonstrating protective potential against oxidative damage and antioxidant action [109]. Furthermore, the hydroalcoholic extract of açaí seeds also reduced lipid peroxidation and protein carbonylation. It increased the antioxidant activity of superoxide dismutase, catalase, and glutathione peroxidase in the brainstem of adult offspring subjected to chronic maternal separation (Table 1 and Figure 4) [89].

The molecular mechanisms of this antioxidant property of açaí involve the transcription of the nuclear factor erythroid 2-related factor 2 (NRF2), which protects against oxidation of astrocytes and neurons and modulates microglial dynamics [118,119]. Previous studies showed that *Euterpe oleracea* pulp supplementation increased the NRF2 expression in the hippocampus and prefrontal cortex of aged rats [107] and in the hippocampus of rats in a model of vascular dementia [106]. Moreover, heme oxygenase 1 (HO-1) is an enzyme that converts heme with pro-oxidant action into biliverdin and bilirubin, antioxidants that restore the redox state and act beneficially in AD [120]. Açaí pulp also increased HO-1 hippocampal expression in rats with vascular dementia [106]. Finally, *Euterpe oleracea* pulp reduced NADPH-oxidoreductase-2 (NOX-2) expression in the hippocampus of the old rats (Table 1 and Figure 4) [107]. This enzyme modulates anion superoxide mitochondrial production, and its overexpression in the hippocampus impairs cognition [121]. These data highlight açaí’s neuroprotective potential since inflammation and oxidative stress are closely related and may cause memory deficits by compromising hippocampal synaptic plasticity [122].

### 3.3. Euterpe oleracea on Neurogenegis

Studies suggest that the impairment of neurogenesis in the hippocampus may be a critical event in the development of AD. Hippocampal neurogenesis is essential for network maintenance and structural neuronal plasticity [123,124]. In this context, the neurotrophin brain-derived neurotrophic factor (BDNF), which acts by activating its receptor tropomyosin receptor kinase B (TRKB), plays a prominent role in synaptic plasticity, NO production, and long-term potentiation, as well as in the modulation of neuronal survival and differentiation [125]. Moreover, this neurotrophin also causes tau dephosphorylation [126]. Therefore, changes that reduce BDNF levels, such as AD development, impact hippocampal function and memory [127,128,129].

Notably, the hydroalcoholic extract of açaí seeds activates the NO-BDNF-TRKB pathway since it normalizes NO levels and increases the expression of the TRKB receptor in the hippocampus of adult pups subjected to chronic maternal separation (Table 1 and Figure 4) [89]. Previous studies suggest that substances that target the BDNF pathway have beneficial therapeutic potential for acting on cognition [130,131,132]. Therefore, açaí is available as a promising natural product for preventing and treating cognitive deficits in AD.

### 3.4. Euterpe oleracea on Apoptosis

AD progression is associated with the loss of connections between brain cells, resulting in cell death and worsening cognitive symptoms. Neurodegeneration occurs primarily in the entorhinal cortex, the hippocampal formation, and the association regions of the neocortex [133]. In this neurodegenerative disease, the progressive loss of neurons involves oxidative stress associated with mitochondrial dysfunction, inflammation, gliosis, axonal degeneration, and the impairment of synaptic transmission [133,134,135]. Moreover, studies suggest that activated caspase-3 is essential in the progressive loss of neurons associated with the disease [136,137].

Açaí pulp increases the antiapoptotic B-cell lymphoma 2 (BCL-2) RNAm expression and reduces the pro-apoptotic BCL-2-associated X protein (BAX) expression in rats with vascular dementia (Table 1 and Figure 4) [106]. These alterations play an essential role in apoptosis since increased intracellular BAX and reduced BCL-2 lead to reduced mitochondrial membrane permeability, promoting the release of cytochrome C into the cell plasma. Thus, they activate caspase 9, the activator, leading to the formation of the apoptotic complex. Subsequently, caspase-3, or effector caspase, is activated, promoting cell death [138]. Additionally, caspase-3 also acts in the cleavage of tau protein and APP, contributing to the formation of beta-amyloid plaques in the brains of patients. Therefore, drugs capable of preventing the activation and execution of apoptosis by caspase-3 represent a promising approach in the treatment of Alzheimer’s disease [136,139], which highlights the pharmacological potential of açaí.

### 3.5. Euterpe oleracea on Autophagy

Autophagy is a catabolic process in which cells digest constituents of the cytoplasm, such as dysfunctional organelles and misfolded proteins [140]. The literature data describe three main types of autophagy: chaperone-mediated autophagy (CMA), microautophagy, and macroautophagy [141]. Lysosomes are cell organelles that degrade and recycle cellular waste and fuse with autophagosomes. Subsequently, proteolytic lysosomal enzymes perform substrate degradation, while vesicular or vacuolar ATPase (VATPase) mediates acidification of the compartment [142]. Therefore, autophagy is a fundamental process for neurons to eliminate large insoluble protein aggregates, which become vulnerable when dysfunctional [143]. In addition, there is a small distribution of lysosomes in the distal axons, so autophagosomes must be transported to the cell body [144,145].

Previous studies suggest that impaired autophagy contributes to the pathogenesis of AD and other neurodegenerative diseases [146,147,148]. There are also reports of the accumulation of immature autophagosomes in the brains of patients with this disease, the downregulation of autophagy-related proteins [149], and the accumulation of autolysosomal vesicles in axons, promoting network impairment and AD progression [150]. Pretreatment with the extract obtained from açaí pulp caused the clearance of autophagic vacuoles in cultures of HT22 hippocampal neuron cells subjected to bafilomycin A1, an autophagy inhibitor. Açaí also reduced the ratio of LC3-II to LC3-I in these cells, indicating the occurrence of a rapid turnover of vacuolar structures since this protein facilitates the fusion and renewal of damaged proteins and organelles, encapsulated in autophagic vacuoles bound for lysosomes. Additionally, *Euterpe oleracea* pulp reduced the mechanistic target of rapamycin (mTOR) phosphorylation, which markedly increased autophagy and decreased the accumulation of p62/SQSTM1, known as sequestosome 1, in this cell culture (Table 1 and Figure 4) [110].

Preclinical studies have also investigated the effects of açaí on autophagy in animal models. The supplementation of aged animals with açaí pulp increased the expression of Beclin-1 in the prefrontal cortex, a protein that plays a critical role in initiating autophagy. In addition, *Euterpe oleracea* pulp also inhibited the accumulation of sequestosome 1 (p62/SQSTM1) in the prefrontal cortex and reduced the ratio of MAP1B-LC3II to LC3I in the hippocampus and prefrontal cortex of these aged rats [107]. Another research group demonstrated that açaí pulp increased the mRNA expression of protein that in humans is encoded by the BECN1 gene (Beclin-1) and reduced LC3B and p62 in the hippocampus of rats subjected to the vascular dementia model (Table 1 and Figure 4) [106].

It is worth noting that inhibiting mTOR-dependent mechanisms increases autophagy and reduces the deposition of intracellular beta-amyloid protein in the brain [151,152], as well as the hyperphosphorylation of TAU [153]. Furthermore, the activation of Beclin-1 reverses cognitive deficits and beta-amyloid protein deposition [154,155]. Therefore, these beneficial effects of açaí on autophagy highlight its potential for preventing and treating AD.

### 3.6. Pharmacokinetic Characteristics and Potential Therapeutic Efficacy in Humans

Flavonoids, such as catechins, epicatechins, and anthocyanins present in açaí, are absorbed and metabolized as glycosides and aglycones, mainly in the small intestine [156]. Meanwhile, polymeric proanthocyanidins present in the açaí seed do not appear to be absorbed in the small intestine [157]. The membrane transporters, sodium-dependent glucose transporter 1 (SGLT-1) and glucose transporter 2 (GLUT-2), transport flavonoids into the cytoplasm, where the enzymes β-glucosidase (β-Gluc) or lactase phlorizin hydrolase (LPH) hydrolyze the flavonoid glycosides into their respective aglycones for absorption. These compounds undergo hepatic metabolism by phase I and II reactions, such as hydroxylation, glucuronidation, sulfation, and methylation, for their elimination in urine and bile [156]. Moreover, data from the literature has also demonstrated that the polyphenols present in açaí seeds maintain their antioxidant scavenging capacity and anti-inflammatory properties after gastrointestinal digestion, suggesting their pharmacological potential [158].

Substances that act on the central nervous system must cross the blood–brain barrier (BBB) to exert their effects. The BBB is a structure that regulates the transport of substances between the blood and the central nervous system. It has occluded zonules, forming an intact membrane that seals the spaces between cells [159]. In this context, the pulp and seed of the açaí berry are rich in polyphenols, which need to cross the BBB to exert their effects on the central nervous system. A recent study quantified the polyphenols and their metabolite content in the cerebrospinal fluid of 90 individuals at risk of developing dementia using chromatography–mass spectrometry. This analysis revealed that polyphenols can cross the BBB through passive diffusion or by utilizing transporters, thereby promoting neuroprotective effects [160].

Although there are no clinical studies using extracts prepared from the pulp and seeds of the açaí berry on cognition, data from the literature investigated the effects of anthocyanin administration in patients at increased risk of developing dementia, in which they used 320 mg of anthocyanins per day [161,162], which could be a starting point for transposing the concentrations used in the extracts of the pulp and seed of açaí in preclinical studies to investigations in humans. It is worth noting that hydroalcoholic extracts have the potential to enhance the extraction of polyphenols present in the pulp and seeds, thereby differentiating their administration from traditional fruit consumption, which is in agreement with data from the literature that demonstrated that the ethanolic extract of the pulp and seed of açaí had more prominent pharmacological effects [28,101]. In addition, our research group emphasizes the importance of the synergistic effect of the bioactive compounds present in the extracts, which may differ from the actions of administering the isolated compound. Finally, some techniques can provide new formulations with enhanced bioavailability, thereby increasing the ability of the extracts to promote their neuroprotective properties, such as nanoemulsions and ultrasonication processes, among others.

It is also worth highlighting the possibility of using the pulp and seed of the açaí berry as dietary supplements or nutraceuticals as a preventive measure or during the treatment of neurotoxicity and neurodegeneration and the cognitive deficits associated with them. The flour derived from the açaí seed, used in the local cuisine of Pará, has pharmacological potential [163]. Moreover, a previous study demonstrated that the aqueous and hydroalcoholic extracts prepared from açaí berry pulp contain nutraceutical components with anticholinesterase and antioxidant capabilities [101], offering a beneficial dietary component that limits the pathological deficits characteristic of AD.

## 4. Conclusions

The reviewed preclinical studies demonstrate that açaí pulp prevents and reverses cognitive and memory deficits in different experimental models, highlighting its therapeutic potential for AD patients’ primary symptoms. Regarding this beneficial effect, the fruit’s pulp comprises phenolic compounds that play a fundamental role, as they are potent antioxidant agents. Polyphenols can act directly in the neutralization of reactive species by donating electrons or indirectly by increasing the synthesis or activity of antioxidant enzymes, such as superoxide dismutase, catalase, and glutathione peroxidase, thereby reducing the neurotoxicity present in the pathophysiology of the disease. In addition, the anti-inflammatory action of these compounds present in the pulp of *Euterpe oleracea* minimizes the activation of microglia and the release of pro-inflammatory cytokines, contributing to the reduction in oxidative stress and the accumulation of beta-amyloid plaques. Notably, açaí pulp also acts by inhibiting apoptosis and restoring autophagy, mechanisms that play a prominent role in reducing neurodegeneration and contribute to its therapeutic action in AD.

Due to the close relationship between aging and oxidative stress, the antioxidant property is the primary beneficial neuroprotective effect of açaí pulp and seeds. The antioxidant action caused by its bioactive compounds is directly related to its anti-inflammatory action, which in turn reduces neurotoxicity and neurodegeneration, improving cognition. Therefore, the potent antioxidant effect of *Euterpe oleracea* is involved with all other mechanisms of action described in this review.

Regarding the properties of the açaí seed on the central nervous system, we still lack studies that elucidate its role in cognitive and memory deficits. However, the fruit seed is rich in polyphenols, has central antioxidant action, and stimulates hippocampal neurogenesis, a promising action for treating AD. Notably, the seed represents the fruit’s most significant part, usually discarded after the pulp is collected. Thus, its pharmacological potential may provide a purpose for it.

Furthermore, the development of new pharmaceutical formulations with the potential to increase the bioavailability of açaí’s bioactive compounds may enhance their effects on the central nervous system. Another concern is the half-life, volume of distribution, and other pharmacokinetic characteristics of açaí that require further study to determine the appropriate dosage interval for administration.

Although there are few studies on the effects of *Euterpe oleracea* in experimental models that mimic AD, this review highlights its promising role since this medicinal plant demonstrates action on the main pathophysiological alterations of AD, unlike what occurs with available pharmacological therapies, which may favor its therapeutic potential for the prevention and treatment of this neurodegenerative disease. However, more preclinical studies are needed, mainly with the use of açaí seeds, to deepen our knowledge about its mechanisms of action and pharmacokinetic characteristics for conducting clinical studies in the future.

## Figures and Tables

**Figure 1 plants-14-02010-f001:**
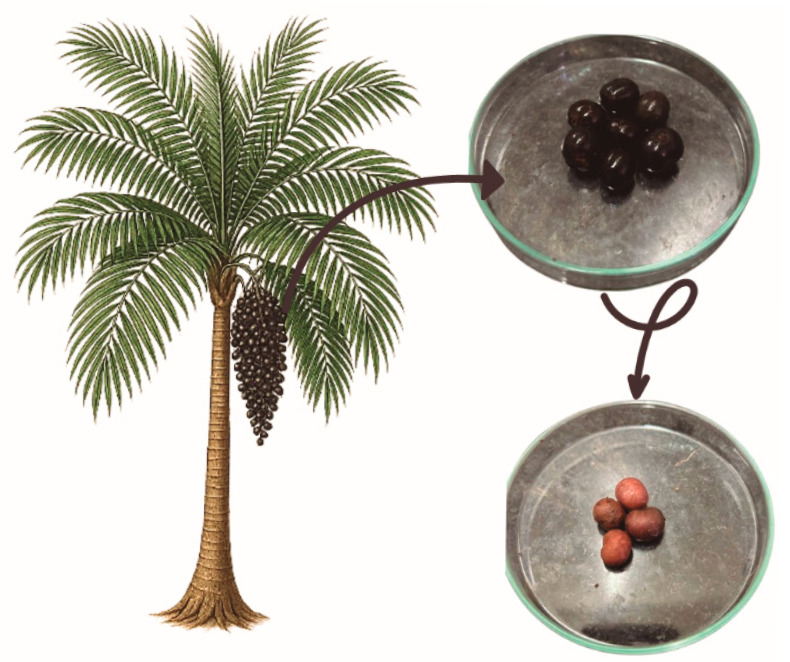
The palm tree *Euterpe oleracea Martius* illustration shows its fruits, including photos of the açaí with the pulp (in the Petri dish at the top) and the açaí seed (in the Petri dish at the bottom).

**Figure 2 plants-14-02010-f002:**
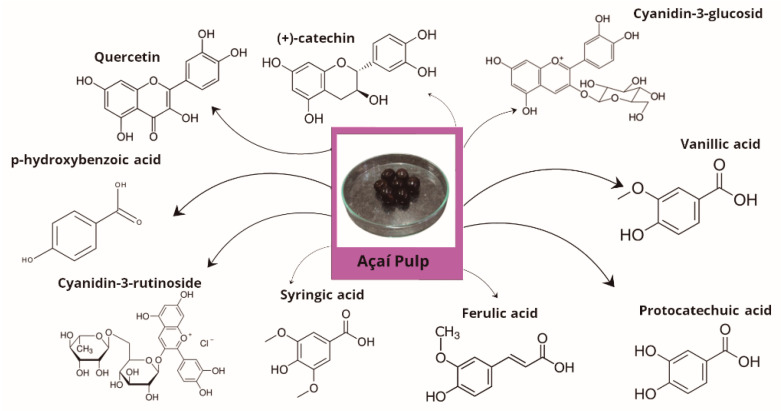
Major phytochemical compounds are present in açaí pulp composition.

**Figure 3 plants-14-02010-f003:**
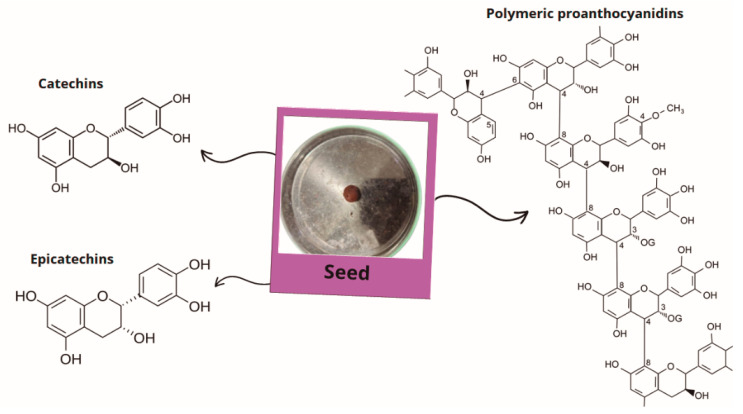
Major phytochemical compounds are present in açaí seed composition.

**Figure 4 plants-14-02010-f004:**
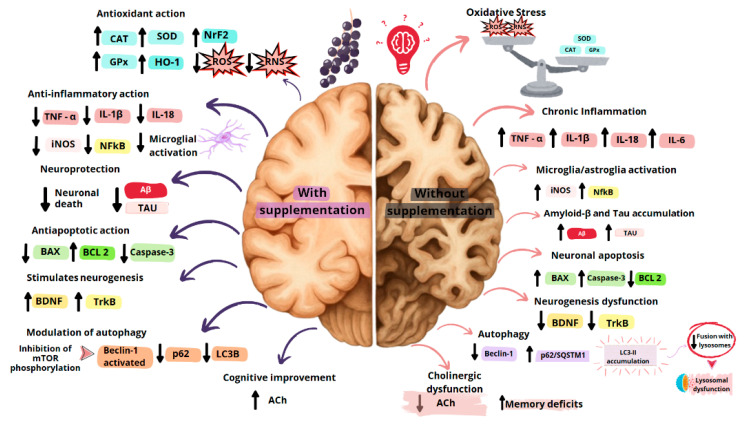
The therapeutic effects of *Euterpe oleracea* pulp and seeds on learning and memory, antioxidants, anti-inflammatory, and antiapoptotic activity, neurogenesis, and autophagy restoration. Abbreviations: SOD, superoxide dismutase; CAT, catalase; GPx, glutathione peroxidase; NRF2, nuclear factor erythroid 2-related factor 2; ROS, reactive oxygen species; RNS, reactive nitrogen species; TNF-α, tumor necrosis factor alpha; IL-1β, interleukin 1 beta β; IL-18, interleukin 18; iNOS, inducible nitric oxide synthase; NFkB, nuclear factor kappa B; BAX, pro-apoptotic BCL-2-associated X protein; BCL-2, antiapoptotic B-cell lymphoma 2; BDNF, brain-derived neurotrophic factor; TRKB, tropomyosin receptor kinase B; mTOR, mechanistic target of rapamycin; Beclin 1, protein that in humans is encoded by the BECN1 gene; p62, sequestosome 1; LC3B, microtubule-associated protein 1A/1B-light chain 3.

**Table 1 plants-14-02010-t001:** Therapeutic effects of *Euterpe oleracea* pulp and seed in the central nervous system.

Experimental Model	Treatment	Mechanisms and Results	References
Male rats submitted to scopolamine and mecamylamine	Açaí pulp 100 and 300 mg/kg	Improved cognition and increased hippocampal acetylcholine	[95]
In vitro	Açaí pulp 0.001 at 1000 µg/mL	Inhibited acetylcholinesterase and butyrylcholinesterase activity	[101]
Male old rats and BV-2 cells	Açaí pulp 2%	Improved cognition, reduced microglial activation and NO levels	[104]
Male obese mice	Açaí pulp 2%	Improved cognition, increased insulin sensitivity, adiponectin levels, and antioxidant activity	[105]
Male mice with vascular dementia	Açaí pulp 500 mg/kg	Improved cognition, reduced apoptosis, restored autophagy, and increased antioxidant activity in the hippocampus	[106]
BV-2 cells submitted to LPS	Açaí pulp 50, 125, 250, 500, and 1000 µg/mL	Reduced NO, iNOS, COX-2, TNF-α, and NFkB	[40]
Male old rats	Açaí pulp 2%	Reduced NFkB and NOX-2 in the hippocampus. Increased NRF2 in the hippocampus and prefrontal cortex. Elevated Beclin 1 expression in the prefrontal cortex	[107]
Male rats submitted to CCl4	Açaí pulp 7 μL/g	Reduced TNF-α, IL-1β, IL-18, and oxidative stress in the cerebral cortex, cerebellum, and hippocampus	[108]
Cerebral cortex, cerebellum, and hippocampus homogenates from rats submitted to H_2_O_2_	Açaí pulp 40% wt/vol	Reduced lipid peroxidation and protein carbonilation, increased SOD and CAT activity	[109]
Adult male offspring subjected to chronic maternal separation	Açaí seed extract 200 mg/kg	Reduced lipid peroxidation and protein carbonilation, increased SOD, GPx, and CAT activity in the brainstem. Normalized NO levels and increased TRKB expression in the hippocampus	[89]
HT22 hippocampal cells	Açaí pulp 0.25 to 1 mg/mL	Restored autophagy	[110]
Human red blood cells submitted to D-galactose	Açaí pulp 10 µg/mL	Mitigated cell deformation	[111]
Human erythrocytes submitted to D-galactose	Açaí pulp 0.5 and 10 µg/mL	Prevented the formation of acanthocytes and leptocytes	[112]
Microglia EOC cell line submitted to LPS and Nigericin	Açaí pulp 500 and 1000 μg/mL	Inhibited NLRP3 inflammasome and reduced IL-1β	[113]
In silico	Açaí pulp	Inhibited NLRP3 inflammasome and highlighted the low toxicity of açaí compounds	[114]

Abbreviations: BV-2 cells, microglial cells derived from C57/BL6 murine; NO, nitric oxide; LPS, lipopolysaccharides; iNOS, inducible nitric oxide synthase; COX-2, cyclooxygenase-2; TNF-α, tumor necrosis factor alpha; NFkB, nuclear factor kappa B; CCl_4_, carbon tetrachloride; NOX-2, NADPH-oxidoreductase-2; IL-1β, interleukin 1 beta β; IL-18, interleukin 18; H_2_O_2_, hydrogen peroxide; SOD, superoxide dismutase; CAT, catalase; GPx, glutathione peroxidase; TRKB, tropomyosin receptor kinase B; HT22 cells, cell line derived from primary mouse hippocampal neurons; Beclin 1, protein that in humans is encoded by the BECN1 gene.

## Data Availability

Not applicable.

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
