# Peer review of "Antioxidant, Anti-Inflammatory, and Antiapoptotic Effects of *Euterpe oleracea* Mart. (Açaí) in Improving Cognition Deficits: Potential Therapeutic Implications for Alzheimer’s Disease"

_plants, 2025, doi:10.3390/plants14132010_

Round 1

Reviewer 1 Report

Comments and Suggestions for Authors

This manuscript presents a narrative review of the potential neuroprotective effects of Euterpe oleracea pulp and seed, particularly in the context of Alzheimer's disease (AD). The topic is both timely and highly relevant, given the increasing global burden of neurodegenerative diseases and the growing interest in plant-based therapeutic strategies. While the manuscript is well-structured and encompasses a wide range of preclinical findings, several critical areas require substantial improvement to strengthen its scientific rigor.

  1. The section detailing the role of açaí seed in the central nervous system (CNS) and Alzheimer's disease (AD) is notably underdeveloped and lacks substantial support from current scientific evidence.
  2. While the manuscript effectively reviews numerous preclinical findings, it lacks a critical discussion regarding the challenges of translating these results to human applications. Key considerations such as polyphenol bioavailability, blood-brain barrier (BBB) permeability, dose extrapolation, and pharmacokinetics are notably absent. Addressing these factors is crucial for a comprehensive understanding of açaí's potential therapeutic efficacy in humans.
  3. The manuscript discusses multiple mechanisms (NRF2, BDNF, mTOR, etc.), but lacks an integrative model or hierarchy of key targets.
  4. Most of the cited experimental studies are relatively outdated (prior to 2020), and the manuscript does not incorporate recent clinical trials, systematic reviews, or newly identified mechanisms. Updating the reference list to include more recent studies from 2022 to 2024 would significantly enhance the manuscript’s relevance, scientific rigor, and credibility.
  5. Ensure consistent use of abbreviations throughout the manuscript (e.g., TNF-α, NOX-2), and provide clear definitions upon their first appearance. Additionally, carefully review the text for grammatical accuracy and minimize the excessive use of passive voice to improve clarity and readability.

Author Response

To Reviewer #1 comments:

The authors thank the reviewer for his/her time and valuable suggestions. We have addressed the comments of this reviewer as follows:

This manuscript presents a narrative review of the potential neuroprotective effects of Euterpe oleracea pulp and seed, particularly in the context of Alzheimer's disease (AD). The topic is both timely and highly relevant, given the increasing global burden of neurodegenerative diseases and the growing interest in plant-based therapeutic strategies. While the manuscript is well-structured and encompasses a wide range of preclinical findings, several critical areas require substantial improvement to strengthen its scientific rigor.

  1. Reviewer comment: The section detailing the role of açaí seed in the central nervous system (CNS) and Alzheimer's disease (AD) is notably underdeveloped and lacks substantial support from current scientific evidence.

Authors’ response: Thank you very much for your valuable observation. Our research group is at the forefront of investigating the effects of açaí seeds on the central nervous system, a novel area of study. We have demonstrated its promising central and peripheral effects, and I did not find any studies by other research groups in the literature that investigate the effects of açaí seeds that could contribute to this review.

We decided to include these data on the açaí seed in the review due to its innovative potential, as it utilizes a part of the fruit that is typically discarded, which has a high polyphenolic content that is more prominent than that of the pulp and exhibits relevant actions on the central nervous system, including antioxidant and neurogenesis effects. We are currently investigating the effects of the hydroalcoholic extract of the açaí seed in an experimental model of schizophrenia, but the data are still preliminary.

  1. Reviewer comment: While the manuscript effectively reviews numerous preclinical findings, it lacks a critical discussion regarding the challenges of translating these results to human applications. Key considerations such as polyphenol bioavailability, blood-brain barrier (BBB) permeability, dose extrapolation, and pharmacokinetics are notably absent. Addressing these factors is crucial for a comprehensive understanding of açaí's potential therapeutic efficacy in humans.

Authors’ Response:  We agree with the reviewer that these are critical points for understanding the therapeutic efficacy of açaí in humans. Thank you very much for highlighting the importance of discussing these points in the review. As recommended by the reviewer, we have included the discussion of these points as follows:

3.5. Pharmacokinetic Characteristics and Potential Therapeutic Efficacy in Humans

Flavonoids, such as catechins, epicatechins, and anthocyanins present in açaí, are absorbed and metabolized as glycosides and aglycones, mainly in the small intestine (Hu et al. Molecules. 2025; 30: 1184). Meanwhile, polymeric proanthocyanidins present in the açaí seed do not appear to be absorbed in the small intestine (Donovan et al. Plant Secondary Metabolites; JohnWiley & Sons, Inc.: Hoboken, NJ, USA, 2006; pp. 303–351). The membrane transporters sodium-dependent glucose transporter 1 (SGLT-1) and glucose transporter 2 (GLUT-2) transport flavonoids into the cytoplasm, where the enzymes β-glucosidase (β-Gluc) or lactase phlorizin hydrolase (LPH) hydrolyze the flavonoid glycosides into their respective aglycones for absorption. These compounds undergo hepatic metabolism by phase I and II reactions, such as hydroxylation, glucuronidation, sulfation, and methylation, for their elimination in urine and bile (Hu et al. Molecules. 2025; 30: 1184). Moreover, data from the literature has also demonstrated that the polyphenols present in açaí seeds maintain their antioxidant scavenging capacity and anti-inflammatory properties after gastrointestinal digestion, suggesting their pharmacological potential (Melo et al. Heliyon. 2020; 6: e05214).

Substances that act on the central nervous system must cross the blood-brain barrier (BBB) to exert their effects. The BBB is a structure that regulates the transport of substances between the blood and the central nervous system. It has occluded zonules, forming an intact membrane that seals the spaces between cells (Alahmari. Neural Plast. 2021; 2021: 6564585). In this context, the pulp and seed of the açaí berry are rich in polyphenols, which need to cross the BBB to exert their effects on the central nervous system. A recent study quantified the polyphenol and their metabolite content in the cerebrospinal fluid of 90 individuals at risk of developing dementia using chromatography-mass spectrometry. This analysis revealed that polyphenols can cross the BBB through passive diffusion or by utilizing transporters, thereby promoting neuroprotective effects (Le Sayec et al. Food Funct. 2023; 14: 8893–8902).

Although there are no clinical studies using extracts prepared from the pulp and seeds of the açaí berry on cognition, data from the literature investigated the effects of anthocyanin administration in patients at increased risk of developing dementia, in which they used 320 mg of anthocyanins per day (Aarsland et al. Am J Geriatr Psychiatry. 2023 Feb;31: 141-151; Khalifa et al. Front Neurol. 2020; 11: 916), which could be a starting point for transposing the concentrations used in the extracts of the pulp and seed of açaí in preclinical studies to investigations in humans. It is worth noting that hydroalcoholic extracts have the potential to enhance the extraction of polyphenols present in the pulp and seeds, thereby differentiating their administration from traditional fruit consumption, which is in agreement with data from the literature that demonstrated that the ethanolic extract of the pulp and seed of the açaí had more prominent pharmacological effects (AlNasser et al. Molecules. 2022; 27: 4891; Rocha et al. Vascul Pharmacol. 2007; 46: 97-104). In addition, our research group emphasizes the importance of the synergistic effect of the bioactive compounds present in the extracts, which may differ from the actions of administering the isolated compound. Finally, some techniques can provide new formulations with enhanced bioavailability, thereby increasing the ability of the extracts to promote their neuroprotective properties, such as nanoemulsions and ultrasonication processes, among others.

  1. Reviewer comment: The manuscript discusses multiple mechanisms (NRF2, BDNF, mTOR, etc.), but lacks an integrative model or hierarchy of key targets.

Authors’ Response: We sincerely appreciate your insightful comment.

Due to the close relationship between aging and oxidative stress, the antioxidant property is the primary beneficial neuroprotective effect of the açaí pulp and seed. The antioxidant action caused by its bioactive compounds is directly related to its anti-inflammatory action, which in turn reduces neurotoxicity and neurodegeneration, improving cognition. Therefore, the potent antioxidant effect of Euterpe oleracea is involved with all other mechanisms of action described in this review.

  1. Reviewer comment: Most of the cited experimental studies are relatively outdated (prior to 2020), and the manuscript does not incorporate recent clinical trials, systematic reviews, or newly identified mechanisms. Updating the reference list to include more recent studies from 2022 to 2024 would significantly enhance the manuscript’s relevance, scientific rigor, and credibility.

Authors’ Response: As recommended by the reviewer, we have updated the reference list to include more recent studies and discussed them in the review.

- Gao et al. Adv Clin Exp Med. 2024; 33: 1179–1187.

- Spinelli et al. Antioxidants. 2023; 12: 848.

- Remigante et al. Cells. 2022; 11: 2391.

- Taneva et al. IJMS. 2023; 24: 14296.

- ALNasser et al. Molecules. 2022; 27: 4891.

- Rocha et al. IJMS. 2024; 25: 8112.

- Aarsland et al. The American Journal of Geriatric Psychiatry. 2023; 31: 141–151.

- Le Sayec et al. Food Funct. 2023; 14: 8893–8902.

- Hu et al. Molecules. 2025; 30: 1184.

- Chen et al. Molecules. 2022; 27: 1816.

Thank you once again for your valuable feedback.

  1. Reviewer comment: Ensure consistent use of abbreviations throughout the manuscript (e.g., TNF-α, NOX-2), and provide clear definitions upon their first appearance. Additionally, carefully review the text for grammatical accuracy and minimize the excessive use of passive voice to improve clarity and readability.

Authors’ Response: As recommended by the reviewer, we carefully reviewed the manuscript text, correcting all acronyms and reducing the use of passive voice.

We sincerely appreciate the valuable comments of the reviewer, which have improved our paper. We have carefully responded to each comment and hope the paper will be suitable for publication in Plants.

Reviewer 2 Report

Comments and Suggestions for Authors

Dear authors, in the attached file are my observations about your submitted article

Author Response

To Reviewer #2 comments:

The authors thank the reviewer for his/her time and valuable suggestions. We have addressed the comments of this reviewer as follows:

You have written a review on the potential properties of Açai berries for the treatment of neurodegenerative diseases, such as Alzheimer disease, based on their polyphenol content.

  1. Reviewer comment: The draft is very well written and easy to read and understand. However, the bibliography is outdated; it uses only one article from 2024 and some from 2023, while the rest are older. When reviewing the databases, there are many later works from 2024 and 2025, which currently renders the manuscript obsolete.

Authors’ Response: We are very grateful for your comment highlighting the need to include studies from 2022 to 2025, which will improve the review. As recommended by the reviewer, we have updated the reference list to include more recent studies and discussed them in the review. However, other articles used in the review, although not as recent, highlight the critical neuroprotective effects of açaí, suggesting its beneficial pharmacological properties on cognition, as well as demonstrating the importance of continuing studies to expand our knowledge about its mechanisms of action, which will inform future clinical studies.

- Gao et al. Adv Clin Exp Med. 2024; 33: 1179–1187.

- Spinelli et al. Antioxidants. 2023; 12: 848.

- Remigante et al. Cells. 2022; 11: 2391.

- Taneva et al. IJMS. 2023; 24: 14296.

- ALNasser et al. Molecules. 2022; 27: 4891.

- Rocha et al. IJMS. 2024; 25: 8112.

- Aarsland et al. The American Journal of Geriatric Psychiatry. 2023; 31: 141–151.

- Le Sayec et al. Food Funct. 2023; 14: 8893–8902.

- Hu et al. Molecules. 2025; 30: 1184.

- Chen et al. Molecules. 2022; 27: 1816.

  1. Reviewer comment: Furthermore, the article only focuses on polyphenols and flavonoids, which are compounds common to many different fruits, such as blueberries, blackberries, grapes, etc., whose high antioxidant power is also proposed as a neuroprotective agent. So, why are acai berries important as a source of pharmacologically active compounds? The authors fail to mention that this fruit has a wide variety of other compounds with potential biological activity, such as amino acid derivatives, fatty acid derivatives, organic acids, etc. that may exert a therapeutic effect and significantly differentiate and highlight Acai from other species with high flavonoid and polyphenol content.

I recommend you study the article: A comparative metabolomics analysis of Acai (Euterpe oleracea Mart.) fruit, food powder, and botanical dietary supplement extracts. Heck KL, Yi Y, Thornton D, Zheng J, Calderón AI. Phytochem Anal. 2025 Mar;36(2):394-408.

Authors’ Response: We understand the reviewer's concern regarding other compounds present in açaí that may also promote beneficial effects. Therefore, we will highlight the reasons for the emphasis on the polyphenols of Euterpe oleracea Martius in their neuroprotective properties.

The reviewer mentions other fruits rich in polyphenols that may also have neuroprotective effects. However, the extent of these effects involves the concentrations of the different types of polyphenols that make up the fruits, as well as their synergistic effects. Most of the studies used in the review investigated the effects of hydroalcoholic extracts of the pulp and seed of the açaí. These extracts have the potential to concentrate polyphenols present in the fruit at higher levels, thereby enhancing their effects (Sasidharan et al. Afr J Tradit Complement Altern Med. 2011; 8: 1-10), which highlights the pharmacological potential of these polyphenols to other bioactive compounds. Furthermore, we believe that the primary action involved in the neuroprotection induced by Euterpe oleracea, as highlighted in the review, is its antioxidant properties. Polyphenols can act directly in the neutralization of reactive species by donating electrons or indirectly by increasing the synthesis or activity of antioxidant enzymes, such as superoxide dismutase, catalase, and glutathione peroxidase, thereby reducing the neurotoxicity present in the pathophysiology of the disease (Krstić et al. Molecules. 2021; 26: 370). Moreover, the antioxidant action caused by its bioactive compounds is directly related to its anti-inflammatory action, which in turn reduces neurotoxicity and neurodegeneration, improving cognition. Therefore, the potent antioxidant effect of Euterpe oleracea is involved with all other mechanisms of action described in this review, highlighting the essential role of the polyphenols.

  1. Reviewer comment: I consider your manuscript to be very incomplete and needs to update and improved. In my opinion, this manuscript, in its current form, should not be considered for publication.

Authors’ Response:  We are very grateful for your comments. To improve our review, we have added more recent articles and a discussion topic on the pharmacokinetic characteristics of bioactive compounds and their potential to cause therapeutic efficacy in humans, as follows:

3.5. Pharmacokinetic Characteristics and Potential Therapeutic Efficacy in Humans

Flavonoids, such as catechins, epicatechins, and anthocyanins present in açaí, are absorbed and metabolized as glycosides and aglycones, mainly in the small intestine (Hu et al. Molecules. 2025; 30: 1184). Meanwhile, polymeric proanthocyanidins present in the açaí seed do not appear to be absorbed in the small intestine (Donovan et al. Plant Secondary Metabolites; JohnWiley & Sons, Inc.: Hoboken, NJ, USA, 2006; pp. 303–351). The membrane transporters sodium-dependent glucose transporter 1 (SGLT-1) and glucose transporter 2 (GLUT-2) transport flavonoids into the cytoplasm, where the enzymes β-glucosidase (β-Gluc) or lactase phlorizin hydrolase (LPH) hydrolyze the flavonoid glycosides into their respective aglycones for absorption. These compounds undergo hepatic metabolism by phase I and II reactions, such as hydroxylation, glucuronidation, sulfation, and methylation, for their elimination in urine and bile (Hu et al. Molecules. 2025; 30: 1184). Moreover, data from the literature has also demonstrated that the polyphenols present in açaí seeds maintain their antioxidant scavenging capacity and anti-inflammatory properties after gastrointestinal digestion, suggesting their pharmacological potential (Melo et al. Heliyon. 2020; 6: e05214).

Substances that act on the central nervous system must cross the blood-brain barrier (BBB) to exert their effects. The BBB is a structure that regulates the transport of substances between the blood and the central nervous system. It has occluded zonules, forming an intact membrane that seals the spaces between cells (Alahmari. Neural Plast. 2021; 2021: 6564585). In this context, the pulp and seed of the açaí berry are rich in polyphenols, which need to cross the BBB to exert their effects on the central nervous system. A recent study quantified the polyphenol and their metabolite content in the cerebrospinal fluid of 90 individuals at risk of developing dementia using chromatography-mass spectrometry. This analysis revealed that polyphenols can cross the BBB through passive diffusion or by utilizing transporters, thereby promoting neuroprotective effects (Le Sayec et al. Food Funct. 2023; 14: 8893–8902).

Although there are no clinical studies using extracts prepared from the pulp and seeds of the açaí berry on cognition, data from the literature investigated the effects of anthocyanin administration in patients at increased risk of developing dementia, in which they used 320 mg of anthocyanins per day (Aarsland et al. Am J Geriatr Psychiatry. 2023 Feb;31: 141-151; Khalifa et al. Front Neurol. 2020; 11: 916), which could be a starting point for transposing the concentrations used in the extracts of the pulp and seed of açaí in preclinical studies to investigations in humans. It is worth noting that hydroalcoholic extracts have the potential to enhance the extraction of polyphenols present in the pulp and seeds, thereby differentiating their administration from traditional fruit consumption, which is in agreement with data from the literature that demonstrated that the ethanolic extract of the pulp and seed of the açaí had more prominent pharmacological effects (AlNasser et al. Molecules. 2022; 27: 4891; Rocha et al. Vascul Pharmacol. 2007; 46: 97-104). In addition, our research group emphasizes the importance of the synergistic effect of the bioactive compounds present in the extracts, which may differ from the actions of administering the isolated compound. Finally, some techniques can provide new formulations with enhanced bioavailability, thereby increasing the ability of the extracts to promote their neuroprotective properties, such as nanoemulsions and ultrasonication processes, among others.

We sincerely appreciate the valuable comments of the reviewer, which have improved our paper. We have carefully responded to each comment and hope the paper will be suitable for publication in Plants.

Reviewer 3 Report

Comments and Suggestions for Authors

Figures 2 and 3. Please maintaining consistent formatting of chemical structures. 

Bioactive compounds, such as polyphenols that are potentially responsible for the health benefits of açaí pulp and seeds have been identified in Section 2, which aligns closely with the manuscript's focus. However, the discussion in this section unnecessarily extends into anti-cancer, anti-diabetic, and hepatoprotective effects of these phytochemicals, which would be better omitted.

Any other bioactive secondary metabolites in açaí pulp and seeds that could be responsible for improving cognition deficits?

Any human intervening studies on the health promoting properties of açaí pulp and seeds have been reported?

I would suggest supplementing discussions regrading the applications of açaí pulp and seeds for functional food formulations and/or nutraceuticals.

The conclusion must identify gaps in research knowledge and offer examples of research that could be done to address those gaps.

Author Response

To Reviewer #3 comments:

The authors thank the reviewer for his/her time and valuable suggestions. We have addressed the comments of this reviewer as follows:

  1. Reviewer comment: Figures 2 and 3. Please maintaining consistent formatting of chemical structures.

Authors’ Response: We greatly appreciate your feedback and the importance you place on the consistent formatting of chemical structures in Figures 2 and 3. As recommended by the reviewer, we improved the chemical structures in the Figures.

  1. Reviewer comment: Bioactive compounds, such as polyphenols that are potentially responsible for the health benefits of açaí pulp and seeds have been identified in Section 2, which aligns closely with the manuscript's focus. However, the discussion in this section unnecessarily extends into anti-cancer, anti-diabetic, and hepatoprotective effects of these phytochemicals, which would be better omitted.

Authors’ Response: We are very grateful for your feedback.

The purpose of this initial section is to demonstrate the well-established pharmacological potential of Euterpe oleracea Martius in peripheral tissues and then to explore further its effects on the central nervous system, which contribute to cognition. These data collectively highlight the promising therapeutic actions of açaí pulp and seeds for the treatment of neurodegeneration, as well as other diseases affecting peripheral tissues. Given the importance of this introduction to highlight the properties of açaí, we have chosen not to remove this text.

  1. Reviewer comment: Any other bioactive secondary metabolites in açaí pulp and seeds that could be responsible for improving cognition deficits?

Authors’ Response: Flavonoids, such as catechins, epicatechins, and anthocyanins present in açaí, are absorbed and metabolized as glycosides and aglycones, mainly in the small intestine (Hu et al. Molecules. 2025; 30: 1184). Meanwhile, polymeric proanthocyanidins present in the açaí seed do not appear to be absorbed in the small intestine (Donovan et al. Plant Secondary Metabolites; JohnWiley & Sons, Inc.: Hoboken, NJ, USA, 2006; pp. 303–351). The membrane transporters sodium-dependent glucose transporter 1 (SGLT-1) and glucose transporter 2 (GLUT-2) transport flavonoids into the cytoplasm, where the enzymes β-glucosidase (β-Gluc) or lactase phlorizin hydrolase (LPH) hydrolyze the flavonoid glycosides into their respective aglycones for absorption. These compounds undergo hepatic metabolism by phase I and II reactions, such as hydroxylation, glucuronidation, sulfation, and methylation, for their elimination in urine and bile (Hu et al. Molecules. 2025; 30: 1184). Moreover, data from the literature has also demonstrated that the polyphenols present in açaí seeds maintain their antioxidant scavenging capacity and anti-inflammatory properties after gastrointestinal digestion, suggesting their pharmacological potential (Melo et al. Heliyon. 2020; 6: e05214).

Substances that act on the central nervous system must cross the blood-brain barrier (BBB) to exert their effects. The BBB is a structure that regulates the transport of substances between the blood and the central nervous system. It has occluded zonules, forming an intact membrane that seals the spaces between cells (Alahmari. Neural Plast. 2021; 2021: 6564585). In this context, the pulp and seed of the açaí berry are rich in polyphenols, which need to cross the BBB to exert their effects on the central nervous system. A recent study quantified the polyphenol and their metabolite content in the cerebrospinal fluid of 90 individuals at risk of developing dementia using chromatography-mass spectrometry. This analysis revealed that polyphenols can cross the BBB through passive diffusion or by utilizing transporters, thereby promoting neuroprotective effects (Le Sayec et al. Food Funct. 2023; 14: 8893–8902).

Thank you once again for your valuable feedback.

  1. Reviewer comment: Any human intervening studies on the health promoting properties of açaí pulp and seeds have been reported?

Authors’ Response: Although there are no clinical studies using extracts prepared from the pulp and seeds of the açaí berry on cognition, data from the literature investigated the effects of an-thocyanin administration in patients at increased risk of developing dementia, in which they used 320 mg of anthocyanins per day [162,163], which could be a starting point for transposing the concentrations used in the extracts of the pulp and seed of açaí in preclin-ical studies to investigations in humans. It is worth noting that hydroalcoholic extracts have the potential to enhance the extraction of polyphenols present in the pulp and seeds, thereby differentiating their administration from traditional fruit consumption, which is in agreement with data from the literature that demonstrated that the ethanolic extract of the pulp and seed of the açaí had more prominent pharmacological effects [28,102]. In addi-tion, our research group emphasizes the importance of the synergistic effect of the bioac-tive compounds present in the extracts, which may differ from the actions of administering the isolated compound. Finally, some techniques can provide new formulations with en-hanced bioavailability, thereby increasing the ability of the extracts to promote their neu-roprotective properties, such as nanoemulsions and ultrasonication processes, among others.

To date, the literature has demonstrated studies in humans investigating the effects of açaí pulp on cardiometabolic diseases, with beneficial properties on these conditions.

Thank you once again for your valuable feedback.

  1. Reviewer comment: I would suggest supplementing discussions regrading the applications of açaí pulp and seeds for functional food formulations and/or nutraceuticals.

Authors’ Response: It is also worth highlighting the possibility of using the pulp and seed of the acai berry as dietary supplements or nutraceuticals as a preventive measure or during the treatment of neurotoxicity, neurodegeneration, and the cognitive deficits associated with them. The flour derived from the açaí seed, used in the local cuisine of Pará, has pharma-cological potential (da Silva et al. Food Res Int. 2018 ; 111: 408-415). Moreover, a previous study demonstrated that the aqueous and hy-droalcoholic extracts prepared from the açaí berry pulp contain nutraceutical components with anticholinesterase and antioxidant capabilities (ALNasser et al. Molecules. 2022; 27: 4891), offering a beneficial dietary component that limits the pathological deficits characteristic of AD.

  1. Reviewer comment: The conclusion must identify gaps in research knowledge and offer examples of research that could be done to address those gaps.

Authors’ Response: As recommended by the reviewer, we have improved the conclusion as follows:

Furthermore, the development of new pharmaceutical formulations with the potential to increase the bioavailability of açaí's bioactive compounds may enhance their effects on the central nervous system. Another concern is the half-life, volume of distribution, and other pharmacokinetic characteristics of açaí that require further study to determine the appropriate dosage interval for administration. 

We sincerely appreciate the valuable comments of the reviewer, which have improved our paper. We have carefully responded to each comment and hope the paper will be suitable for publication in Plants.

Round 2

Reviewer 1 Report

Comments and Suggestions for Authors

accept

Reviewer 3 Report

Comments and Suggestions for Authors

The quality of the revised manuscript has been significantly improved to a state where it can be accepted for publication in plants.